# Changes in gene expression in human skeletal stem cells transduced with constitutively active Gsα correlates with hallmark histopathological changes seen in fibrous dysplastic bone

**Domenico Raimondo[1]☯, Cristina Remoli[1]☯, Letizia Astrologo[2]¤, Romina Burla[2], Mattia La Torre[2], Fiammetta Vernì[2], Enrico Tagliafico[3], Alessandro Corsi[1], Simona Del Giudice[2], Agnese Persichetti[1], Giuseppe Giannicola[5], Pamela G. Robey[4], Mara Riminucci[1]\*, Isabella Saggio[2,6]\***

**1** Department of Molecular Medicine, Sapienza University of Rome, Rome, Italy, **2** Department of Biology and Biotechnology, Sapienza University of Rome, Rome, Italy, **3** Department of Biomedical Sciences, University of Modena and Reggio Emilia, Modena, Italy, **4** National Institute of Dental and Craniofacial Research, NIH, DHHS, Bethesda, MD, United States of America, **5** Department of Anatomical, Histological, Forensic Medicine and Orthopaedics Sciences, Sapienza University of Rome, Rome, Italy, **6** School of Biological Sciences, NTU Institute of Structural Biology, Nanyang Technological University, Singapore

☯ These authors contributed equally to this work.
¤ Current address: Department of Urology and Department for BioMedical Research, Urology Research Laboratory, University of Bern, Bern, Switzerland
\* isabella.saggio@uniroma1.it (IS); mara.riminucci@uniroma1.it (MR)

**Data Availability Statement:** The data discussed in this manuscript have been deposited in NCBI's

## Abstract

Fibrous dysplasia (FD) of bone is a complex disease of the skeleton caused by dominant activating mutations of the *GNAS* locus encoding for the α subunit of the G protein-coupled receptor complex (Gsα). The mutation involves a substitution of arginine at position 201 by histidine or cysteine (Gsα$^{R201H or R201C}$), which leads to overproduction of cAMP. Several signaling pathways are implicated downstream of excess cAMP in the manifestation of disease. However, the pathogenesis of FD remains largely unknown. The overall FD phenotype can be attributed to alterations of skeletal stem/progenitor cells which normally develop into osteogenic or adipogenic cells (*in cis*), and are also known to provide support to angiogenesis, hematopoiesis, and osteoclastogenesis (*in trans*). In order to dissect the molecular pathways rooted in skeletal stem/progenitor cells by FD mutations, we engineered human skeletal stem/progenitor cells with the Gsα$^{R201C}$ mutation and performed transcriptomic analysis. Our data suggest that this FD mutation profoundly alters the properties of skeletal stem/progenitor cells by pushing them towards formation of disorganized bone with a concomitant alteration of adipogenic differentiation. In addition, the mutation creates an altered *in trans* environment that induces neovascularization, cytokine/chemokine changes and osteoclastogenesis. In silico comparison of our data with the signature of FD craniofacial samples highlighted common traits, such as the upregulation of ADAM (A Disintegrin and Metalloprotease) proteins and other matrix-related factors, and of PDE7B (Phosphodiesterase 7B), which can be considered as a buffering process, activated to compensate for

Gene Expression Omnibus and are accessible through GEO Series accession number GSE109818 (https://www.ncbi.nlm.nih.gov/geo/query/acc.cgi?acc=GSE109818). The following secure token has been created to allow review of record GSE109818 while it remains in private status: ctujawowrtehnqf.

**Funding:** This work has been supported by Grants EU FP7 Brainvectors (n. 286071), Telethon GEP15033, PRF 2016-67, Progetti di Ricerca, Sapienza University of Rome (RP1181642E87148C) to IS, FIRC (22392) and CIB to MLT and IS; Avvio alla Ricerca, Sapienza University of Rome (AR2181642B6F2E48, AR1181642EE61111) to RB, SDG and IS; Telethon GGP15198, University of Pennsylvania Orphan Disease Center in partnership with the Fibrous Dysplasia Foundation MDBR16-114-FD/MAS and MDBR17-114-FD/MAS to MR; Sapienza University (RP11715C7C4DC57A) to AC; the IRP, NIH, DHHS (ZIA DE000380) to PGR. The funders had no role in study design, data collection and analysis, decision to publish, or preparation of the manuscript.

**Competing interests:** The authors have declared that no competing interests exist.

excess cAMP. We also observed high levels of CEBPs (CCAAT-Enhancer Binding Proteins) in both data sets, factors related to browning of white fat. This is the first analysis of the reaction of human skeletal stem/progenitor cells to the introduction of the FD mutation and we believe it provides a useful background for further studies on the molecular basis of the disease and for the identification of novel potential therapeutic targets.

## Introduction

Fibrous dysplasia (FD) of bone is a crippling disease of the skeleton that can involve one (monostotic) or several (polyostotic) bones, as an isolated disorder or as a part of the McCune-Albright syndrome (MAS, OMIM #174800). FD/MAS is caused by activating mutations of the *GNAS* locus encoding for the α subunit of the G protein-coupled receptor complex (Gsα) [1–3]. G protein-coupled hormone receptors bind to adenylyl cyclase (AC), necessary for the generation of intracellular cAMP that mediates hormone signaling. In FD/MAS, activating mutation involves a substitution of arginine at position 201 by histidine or cysteine (Gsα$^{R201H}$ or $^{R201C}$) [4]. These mutations inhibit the intrinsic GTPase activity of Gsα, which leads to prolonged stimulation of AC [5]. cAMP causes dissociation of the inactive Protein Kinase A (PKA) tetramer, thereby freeing the catalytic subunits to mediate serine–threonine phosphorylation of target molecules. Several signaling pathways are implicated downstream of excess cAMP in the manifestation of MAS in various tissues [6]. However, the etiology of FD remains largely unknown. In bone, Gsα/cAMP activation increases c-fos expression and this has been demonstrated in FD lesions from patients with MAS [7, 8]. Gsα activity increases the expression of c-fos and other proto-oncogenes through the activation of cAMP-dependent PKA in osteoblastic precursors. Fos binds with jun to form hetero-dimeric complex activator protein 1 (AP-1), which is also highly expressed during the proliferative phase of osteoblast development [1]. Moreover, AP-1 can suppress the expression of late markers of mature osteoblasts, such as osteocalcin [1]. The abnormally differentiated, misfunctioning osteoblasts in FD lesions express elevated levels of IL-6, PDGFβ and sex steroid receptors through a cAMP-dependent mechanism that may be important in osteoclast activation [1]. The increased cAMP level could negatively affect the half-life of Cbfa1/RUNX2 protein, the osteogenic master gene. Changes in expression of these aforementioned genes suggest abnormalities in bone-forming cells, which may contribute to the pattern of inappropriate cell differentiation [3]. In 2010, Kiss et al. examined differential expression of 118 genes in affected versus unaffected human bone tissue of women with FD and they detected marked differences in the transcription profile of 22 genes controlled via G-protein coupled pathways and the BMP cascade, as well as genes coding for extracellular matrix proteins, and in particular, upregulation of a novel gene, *ATP2A2* (*ATPase Sarcoplasmic/Endoplasmic Reticulum Ca2+ Transporting 2*), in FD bone [9]. Along this same line, using microarray analysis, Zhou et al have recently analyzed craniofacial lesional samples from FD patients, identifying *ADAMTS2* (*A Disintegrin and Metalloproteinase with Thrombospondin Type 1 Motif 2*), overexpressed in FD tissues, but rarely expressed in normal bone [10]. Members of the ADAMTS family are involved in controlling extracellular matrix turnover, which in turn, controls several processes including angiogenesis and cell migration. The super-activation of ADAMST2 suggests that extracellular matrix turnover plays a role in FD pathophysiology.

Except for the results described above, no other data are available on the molecular signature of human fibrous dysplastic bone and the manner in which the fundamental effect of

constitutively active Gsα's activity is translated into the distinct histological changes that characterize FD bone and marrow is not well understood. Elucidating the molecular mechanisms of these changes would benefit not only the pursuit of a precise understanding of the when and how a particular FD-unique histological feature develops, but also the quest for therapeutic targets that best relate to the actual, and likely pleiotropic, misfunction of osteogenic cells. These changes, which we have also described through a mouse model representing a direct replica of human FD [11], include: *i*) the replacement of normal marrow tissue with a fibroblastic tissue devoid of hematopoiesis; *ii*) the lack of adipocytes within the affected marrow; *iii*) deposition of excess, structurally abnormal bone; *iv*) defective bone mineralization; *v*) enhanced osteoclastogenesis; and *vi*) abnormal vascularity [11, 12]. In fact, each of these cardinal features can be traced to an alteration of a physiological function served by normal skeletal progenitors, which are known to: *i*) provide a microenvironment for hematopoiesis; *ii*) develop into adipocytes; *iii*) generate osteoblasts; *iv*) produce phosphate-regulating factors; *v*) cue osteoclast progenitors to differentiate into osteoclasts; and *vi*) guide and organize marrow microvessels, with which they physically associate as mural cells [13–15].

Global analysis of the transcriptome and proteome of Gsα-mutated cells is expected to assist in dissecting the diverse downstream effects of constitutively active Gsα on the functions of the relevant cells. In vitro studies conducted so far have capitalized on the use of populations of bone marrow stromal cells derived from the abnormal, fibrotic bone marrow spaces of FD lesions [10]. However, conducting transcriptome or proteome analysis using human clinical material; i.e., a comparison between isolated normal and mutated cells (separated by clonal dilution) has two important limitations: 1) the involvement of multiple variables (patient age, lesion age, anatomical site, type of histopathological changes, concurrent changes in distant, e.g., endocrine, organs affecting bone secondarily) and 2) postulates a large numbers of cells that are not easy to achieve in a rare disease, and independent samples in order to attain statistical significance. Both problems, however, can be circumvented if a reliable way for stable transduction with the causative mutation of the cell type of interest is at hand. We have shown that bone marrow stromal cells (BMSCs, a subset of which are skeletal stem cells, SSCs) can be prospectively isolated from BM by relying on a set of surface markers [13], and that lentiviral technologies can be effectively used to transfer the disease phenotype in these cells [16].

A transduced cell model represents the biological context of choice for studying the acute effects of Gsα mutation, thereby overcoming these limitations. In fact, the strictly paired comparison between mutant cells and their respective controls minimizes the requirements in sample size, and the possibility of describing an effect not directly related to Gsα mutation. Unquestionably, the use of control infection samples is important to identify possible off target or toxic effects of the transduction procedure in this model.

Therefore, one can create stably transduced and reasonably purified populations of skeletal progenitors while retaining untransduced cells from the same donor and tissue sample as controls. By addressing the two major hurdles that stand in the way of conducting high throughput studies using pathological human cells, this approach likely introduces an unrelated bias. Stable ex vivo transduction of skeletal progenitors using lentiviral vectors best portrays effects emerging within a short time frame, affecting a few generation of cells in the lineage, downstream of the transduced progenitor. However, many of the downstream effects that can be envisioned would be less represented. Nonetheless, this can actually be seen as a desirable (albeit incomplete) outcome, as it highlights those cellular changes that drive the *initial* development of tissue changes.

With these aims and limitations in mind, we conducted a global analysis of the transcriptome changes brought about by lentiviral-driven Gsα$^{R201C}$ expression in BMSCs/SSCs using a microarray approach to explore the key molecular events in FD development, and we then

attempted to correlate the results to known cardinal features of FD lesions as defined by histopathology in order to develop potential diagnostic markers or therapeutic targets for FD.

## Results and Discussion

### Gene chip analysis of Gsα$^{R201C}$ hBMSCs

Thus far, the quest for therapeutic approaches, like attempts to interpret the genesis and mechanisms of the disease, have been dominated by almost exclusive consideration of the effect of constitutively active Gsα, with the idea that reverting the effects of mutant Gsα activity in relevant cells would necessarily lead to a cure. However, multiple lines of evidence point to a more complex scenario, in which tissue changes that have a direct bearing on morbidity at the tissue and organ level may well emanate from a complicated and prolonged in vivo history–of the mutated gene, cell, tissue, and organ [11, 17–20].

Keeping these concepts in mind, skeletal stem/progenitor cells isolated from bone marrow of three independent healthy donors and characterized as described previously [13, 21–25] (henceforth referred to as human bone marrow stromal cells, hBMSCs) were stably transduced with a lentiviral vector expressing the constitutively active form of the Gsα protein carrying the R201C mutation (LV-Gsα$^{R201C}$) [16] or with control vector (LV-ctr) or untransduced (mock-treated) controls. Cultures of mock-treated, LV-Gsα$^{R201C}$ and LV-ctr transduced hBMSCs were amplified in basic expansion medium. Ten days following infection, we checked Gsα$^{R201C}$ expression by Western blotting (Fig 1A). Immunoblotting using anti-HA antibody, which highlights the exogenous Gsα$^{R201C}$ expression demonstrates the presence of the mutant protein. We performed in parallel immunoblotting using anti-Gsα antibody showing the presence of a 1.4 fold in average more intense band resulting from the merge of the endogenous Gsα and exogenous Gsα$^{R201C}$ signals. Fifteen days following infection, total RNA was isolated and used for hybridization to an Affymetrix chip to characterize the changes in transcriptomic profile and their relationship to the molecular mechanisms potentially leading to the overall phenotypic changes associated with Gsα$^{R201C}$ mutations in vivo.

Gsα RNA expression 1.6 fold quantification in the infected samples was consistent with immunoblotting data (Fig 1B). Unsupervised hierarchical cluster analysis of differential gene expression of cells from three donors (D01, D02, D03), revealed unique expression profiles associated with the expression of the R201C mutation (R) compared to the mock-treated cells (M) and to the cells transduced with LV-ctr (C) (Fig 1C). Differentially expressed genes were identified by comparing overlapping gene lists obtained using three independent analysis of paired samples as described in detail in the methods section, such as Affymetrix GCOS comparison analysis, dChip Compare Sample procedure, and paired t-test implemented in Partek® GS (S1A Fig). Combining these approaches, 408 genes differentially expressed at ± 2.5-fold levels or greater were identified in comparison to normal and LV-ctr-transduced cells (Figs 1B and 2 and S1 Table). Among the modulated probes, 111 genes were modulated by more than 10-fold (Fig 2), and 95 genes (approximately 23%) were under-expressed, while 313 genes were over-expressed (approximately 76%) (Fig 1D and S1 Table).

To gain more insight into the biological functions of differentially expressed genes, we performed analysis of biological pathways and Gene Ontology (GO) terms associated with the differentially expressed genes by using the list of under- and over-expressed genes as input for the Enrichr tool (http://amp.pharm.mssm.edu/Enrichr/)). Enrichr's Combined Score (ECS), a combination of the p-value and z-score, was used to prioritize the Kyoto Encyclopedia of Genes and Genomes- (KEGG)-enriched pathways and GO results. The most significant KEGG pathways enriched in under-represented genes include *TGFß* (*Transforming Growth Factor-ß*) signaling genes, *WNT* (*Wingless-Type MMTV Integration Site*) signaling genes and

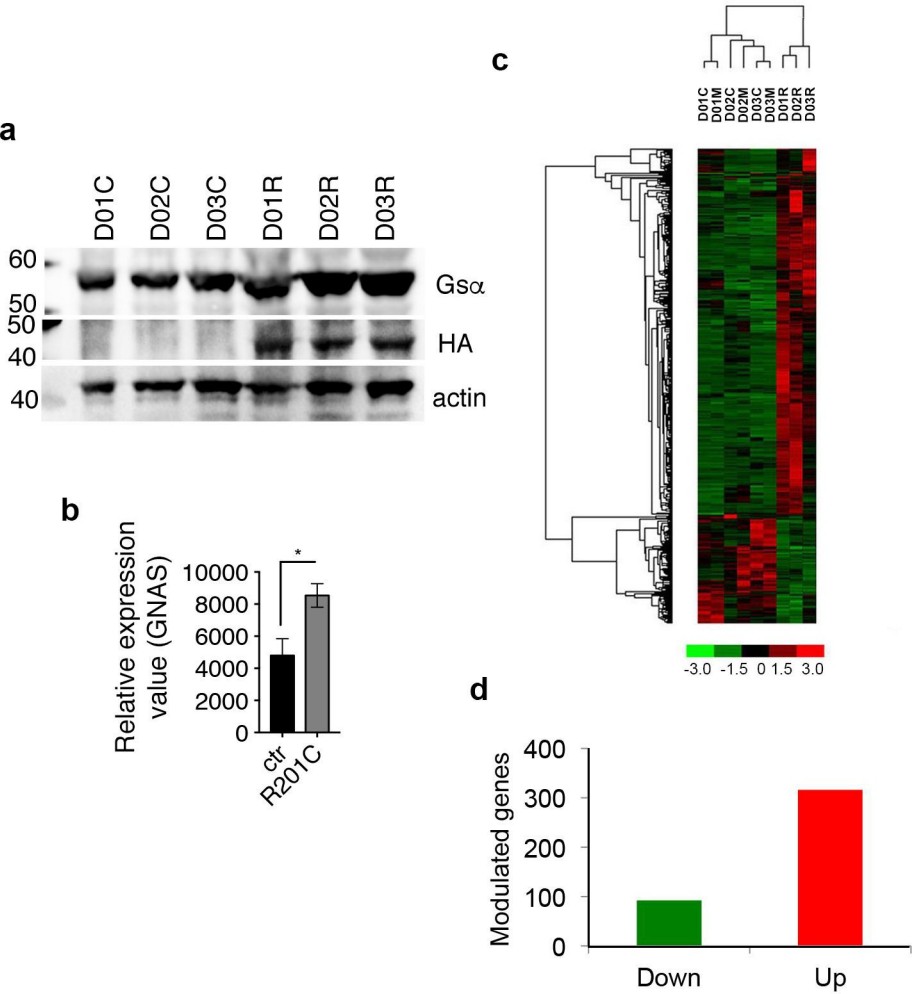

**Fig 1. High throughput GeneChip analysis of Gsα^R201C hBMSCs.** (A) Western blotting on hBMSCs from three different individuals (D01, D02, D03), transduced with LV-Ctr (C), LV- Gsα^R201C (R) showing the expression of the exogenously provided mutant Gsα^R201C as highlighted by immunoblotting with anti-HA and anti-Gsα antibodies. 1.4 fold change on Gsα band in LV- Gsα^R201C (R) respect to that transduced with LV-Ctr (C) has been calculated as the average, on duplicates. (B) Relative values for GNAS expression in D01, D02, D03, transduced with LV-Ctr (ctr) or LV- Gsα^R201C (R201C) showing the expression of the exogenously provided mutant Gsα^R201C. 1.6 fold change on Gsα RNA expression has been calculated as the fold change on the average of triplicate samples from LV- Gsα^R201C and LV-Ctr samples. (C) Hierarchical clustering of modulated genes in triplicate samples of hBMSCs from three different individuals (D01, D02, D03), transduced with LV-Ctr (C), LV- Gsα^R201C (R) or mock-treated (M). (D) Total number of significantly modulated genes (threshold: 2.5). Up-regulated genes are indicated in red, down-regulated in green.

genes involved in osteoclast differentiation (Fig 3A). GO cellular components examined in the context of under-represented genes identified multiple immune cell-related components (Fig 3B). In the same list of significantly under-represented genes, the GO biological processes impacted by the FD mutation showed a significant involvement with categories related to the negative regulation of apoptotic processes and that of cell-cell adhesion (Fig 3C). GO cellular function-enriched results further highlighted that the FD mutation caused epigenetic alterations through the modulation of histone deacetylase activity (Fig 3D). Pathway analysis of the highly over-representative cluster demonstrated enrichment of mostly cell cycle processes, cytokine-cytokine receptor interaction, signaling pathways regulating pluripotency of stem cells, chemokine signaling pathway (Fig 4A). GO analysis indicated the BMP receptor

| HGNC symbol | FC | HGNC symbol | FC |
|---|---|---|---|
| ABL1 | 11.78 | LMO7DN | -11.40 |
| ACAN | -27.11 | LOC440895 | 12.60 |
| ACKR3 | 169.58 | MARCKSL1 | 330.30 |
| ADAM12 | 200.34 | MCC | 158.64 |
| ADAM12 | 12.60 | MID1 | 28.03 |
| ADAMTS5 | 138.84 | MID1 | 27.11 |
| ALG9 | 26.22 | MMP16 | 28.03 |
| ANGPTL2 | 17.58 | PAPPA | 15.91 |
| ANKRD29 | 64.50 | PCDHB10 | 49.40 |
| ANTXR1 | 22.95 | PCDHB2 | 14.39 |
| APELA | 11.40 | PCOLCE2 | 22.20 |
| APOE | 11.40 | PDE4D | 28.98 |
| ARHGAP28 | 13.02 | PDE7B | 665.14 |
| B3GALT2 | -11.02 | PDZRN3 | 78.78 |
| BHLHE40 | 56.45 | PGF | 125.63 |
| BNC2 | 11.40 | PGM2L1 | 18.79 |
| CCL8 | 47.78 | PIK3R3 | 14.39 |
| CHRDL1 | 13.02 | PITPNC1 | 34.24 |
| CLU | 18.79 | PLA2G4A | 37.84 |
| CNTNAP2 | 11.02 | PLSCR1 | 12.60 |
| CRISPLD2 | 14.39 | PLXDC2 | 93.07 |
| CXCL1 | 49.40 | PPARGC1A | 10.66 |
| CXCL12 | 22.95 | RAB20 | 15.91 |
| CXCL13 | 68.95 | RAB27B | 1060.68 |
| CXCL6 | 93.07 | RBP4 | 15.91 |
| DTWD1 | 56.45 | RERG | 54.60 |
| DUSP4 | 21.47 | RNA5SP392 | -19.43 |
| ENOX1 | 78.78 | RORA | 13.46 |
| EPHB1 | 20.77 | SAT1 | 19.43 |
| FAM167A | 18.17 | SCARA5 | 18.17 |
| FAM198B | 252.99 | SERINC5 | 200.34 |
| FAM212B | 28.03 | SFRP4 | 14.39 |
| FAM26F | 46.22 | SLC16A6 | 22.95 |
| FKBP7 | 16.44 | SLC22A23 | 40.45 |
| FLRT2 | 309.00 | SLIT3 | 13.46 |
| FZD1 | 106.34 | SMIM3 | 11.40 |
| GALNT15 | 41.82 | SOBP | 73.70 |
| GEM | 13.46 | SPAG4 | 18.17 |
| GMNN | 20.77 | SPON1 | 22.95 |
| GNAL | 16.44 | SPON1 | 18.17 |
| GNG2 | 10.66 | SPON1 | 12.60 |
| GOLM1 | 30.98 | ST3GAL5 | 10.66 |
| GPR155 | 18.17 | STC1 | 13.02 |
| HEPH | 353.07 | STEAP1 | 21.47 |
| HSPB7 | -10.31 | STEAP4 | 19.43 |
| IGFBP5 | 18.17 | TANC2 | 13.02 |
| INHBE | 40.45 | TF | 13.02 |
| ITGA8 | -20.09 | TIFA | 11.02 |
| ITPRIP | 959.74 | TMEM155 | 93.07 |
| KALRN | 20.77 | TNFAIP6 | 14.88 |
| KCNE4 | 10.31 | TNFSF11 | 75.78 |
| KCNJ8 | 20.77 | VASH2 | 25.36 |
| KIAA1217 | 19.43 | VMO1 | -18.17 |
| KIF26B | 71.28 | WISP1 | 15.38 |
| LAMA4 | 47.78 | WWC1 | 15.91 |
| LINC00473 | 19.43 | | |

**Fig 2. Modulated genes in alphabetical order.** Under-expressed genes (in green) and over-expressed (in red) setting the fold change (FC) threshold at 10. The full list of modulated genes is available in S1 Table.

complex, the positive regulation of cell proliferation and chemokine activity respectively (Fig 4A–4D) as the highest enriched cellular component, biological process and molecular function.

## Bone formation in Gsα<sup>R201C</sup> hBMSCs

In order to relate significantly modulated genes and functional categories to FD pathological traits, we further analyzed the modulated genes by Ingenuity Pathway Analysis (IPA), identified the top ten physiological functions given by IPA core analysis (S1 Fig), and reclustered the identified genes into groups related to fundamental tissue changes that occur in FD, taking into account known hBMSC autocrine and paracrine properties. FD tissue pathology includes abnormal bone formation, adipocyte defects, hematopoiesis defects, undermineralized bone, excess of bone resorption, excess of vascularization [1, 3]. These can be directly (in *cis*) or indirectly (in *trans*) be related to hBMSCs [26], which have the potential of differentiating into chondrocytes, osteoblasts, hematopoiesis-supportive stroma and adipocytes [27], and have paracrine effects on angiogenesis, hematopoiesis and osteoclastogenesis [13, 28, 29].

With this perspective in mind, we observed that the FD mutation affected genes in the WNT signaling pathway, known to regulate bone mass and development [30, 31]. *WNT4* and *WNT5A* were upregulated by the FD mutation, as well as the receptor, *Frizzled Class Receptor 1* (*FZD1*), along with *Secreted Frizzled Related Proteins* (*SFRP1*, *SFRP2* and *SFRP4*), which are thought to be WNT inhibitors. The Gsα<sup>R201C</sup> mutation also modulated matrix and mineralization-related genes including *Matrix Gla Protein* (*MGP*), *Stanniocalcin-1* (*STC1*), *Matrix Metalloproteinases* (*MMPs* such as *MMP2*, *MMP13*) and *A Disintegrin and Metalloproteinase Domain 12* (*ADAM12*) (Fig 5A). MGP is a potent inhibitor of extracellular matrix calcification

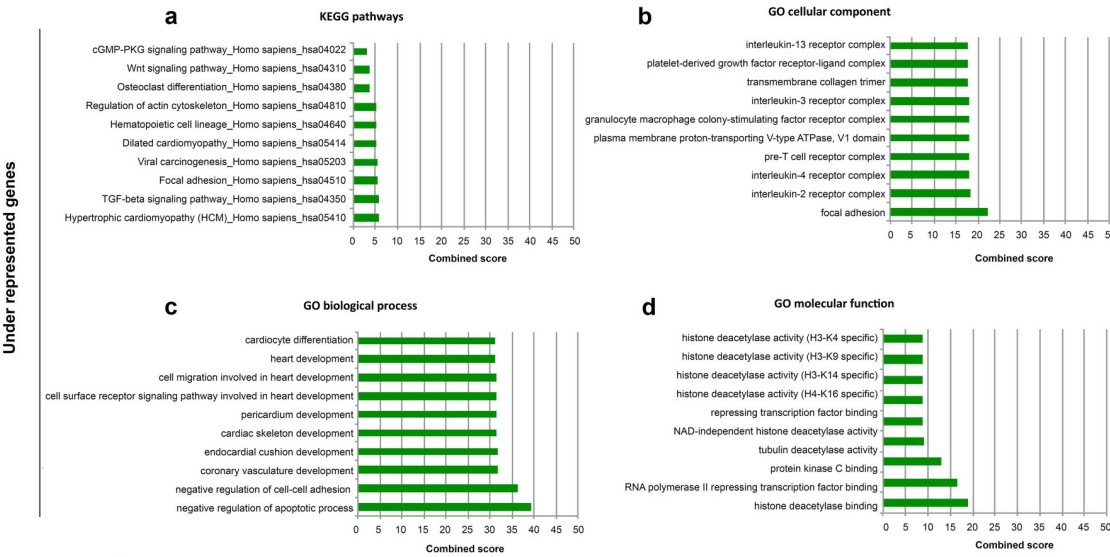

**Fig 3. ENRICHR Gene Ontology and KEGG pathway analysis of significantly under-expressed genes of FD mutated cells.** Enrichment analysis was performed on significantly under-represented genes and ten most significant groups are represented according to KEGG pathway analysis (A), Gene Ontology (GO) cellular component (B), GO biological process (C), and GO molecular function (D). Enricher's combined score combines z-score and p-value.

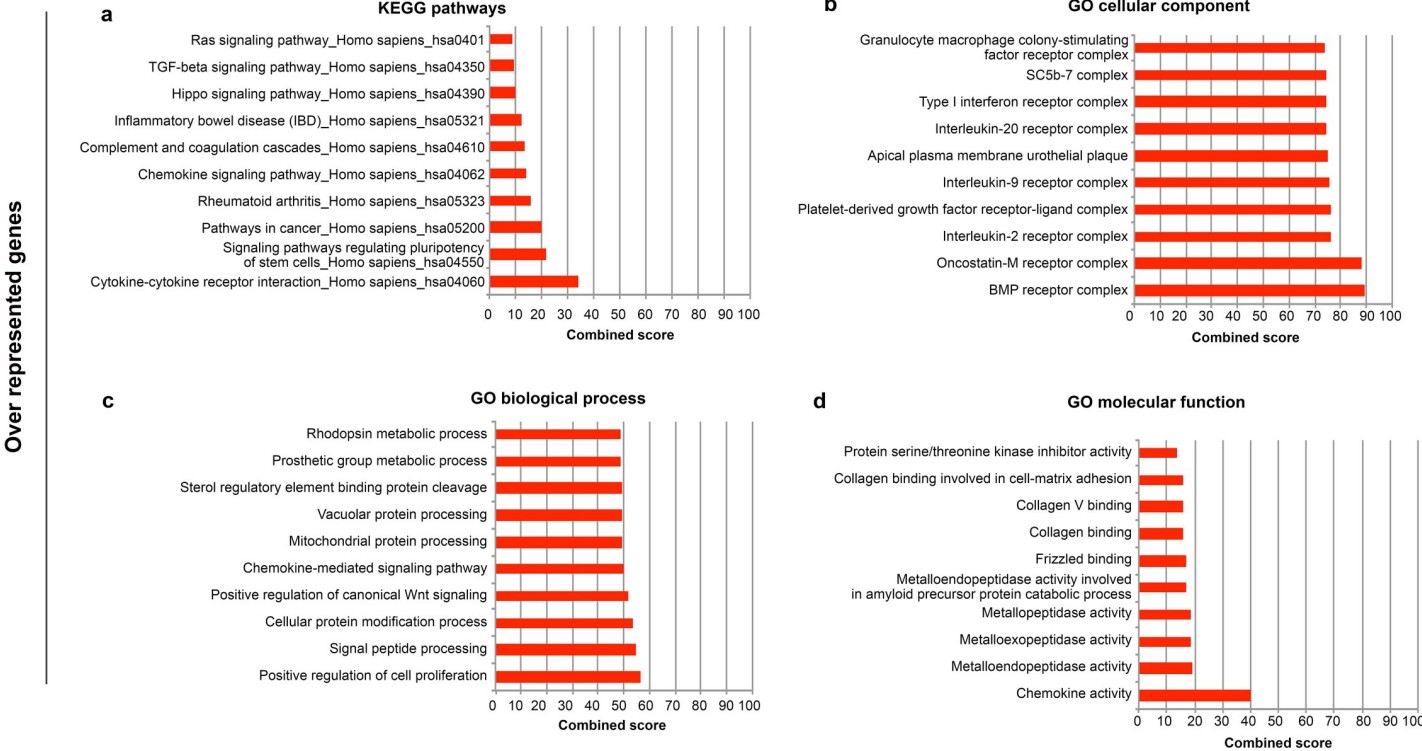

**Fig 4. ENRICHR Gene Ontology and KEGG pathways analysis of significantly over-represented genes of FD mutated cells.** Enrichment analysis was performed on significantly over-represented genes and ten most significant groups are represented according to KEGG pathway analysis (A), Gene Ontology (GO) cellular component (B), GO biological process (C), and GO molecular function (D).

by sequestering $Ca^{2+}$, and transgenic mice overexpressing it in osteoblasts exhibit a reduction in bone mineral levels [32]. STC1, also induced by Gsα[R201C] mutation, is linked with mineralized ossification centers [33], and with calcium and phosphate transport [34]. MMP13, MMP2 and ADAM12 are expressed during skeletal development by chondrocytes, and by synovial cells in arthritis and osteoarthritis [35]. Their activation by the Gsα[R201C] mutation suggests an induction of an intense extracellular remodeling activity in FD [35]. In line with this interpretation, *A Disintegrin and Metalloproteinase Domain with Thrombospondin Type 1 Motif 2* (*ADAMTS2*), was described as an FD biomarker [10], and *MMP2* was defined by Kiss et al as one out of eight upregulated discriminative genes in FD bone tissue [9]. Related to skeletal development, we also observed that *T-Box Protein 3* (*TBX3*) was upregulated. Its function has been linked to stimulation of proliferation of osteogenic precursors and a block of differentiation of cells into mature osteoblasts, which could add a further level of impairment of bone development in FD bone [36].

Taken together, these data indicate that in the initial development of FD lesions, changes in hBMSC transcription induced by the Gsα[R201C] mutation influences bone formation by alteration of WNTs, osteoblast formation and bone matrix remodeling.

## Gsα[R201C] effects on hBMSC adipogenic differentiation

In hBMSCs overexpressing Gsα[R201C], several factors involved in adipogenesis and adipose metabolism were found upregulated (Fig 5B). Among these we found *Lipoprotein Lipase* (*LPL*), an important adipocyte marker implicated in fatty acid accumulation [37]. The

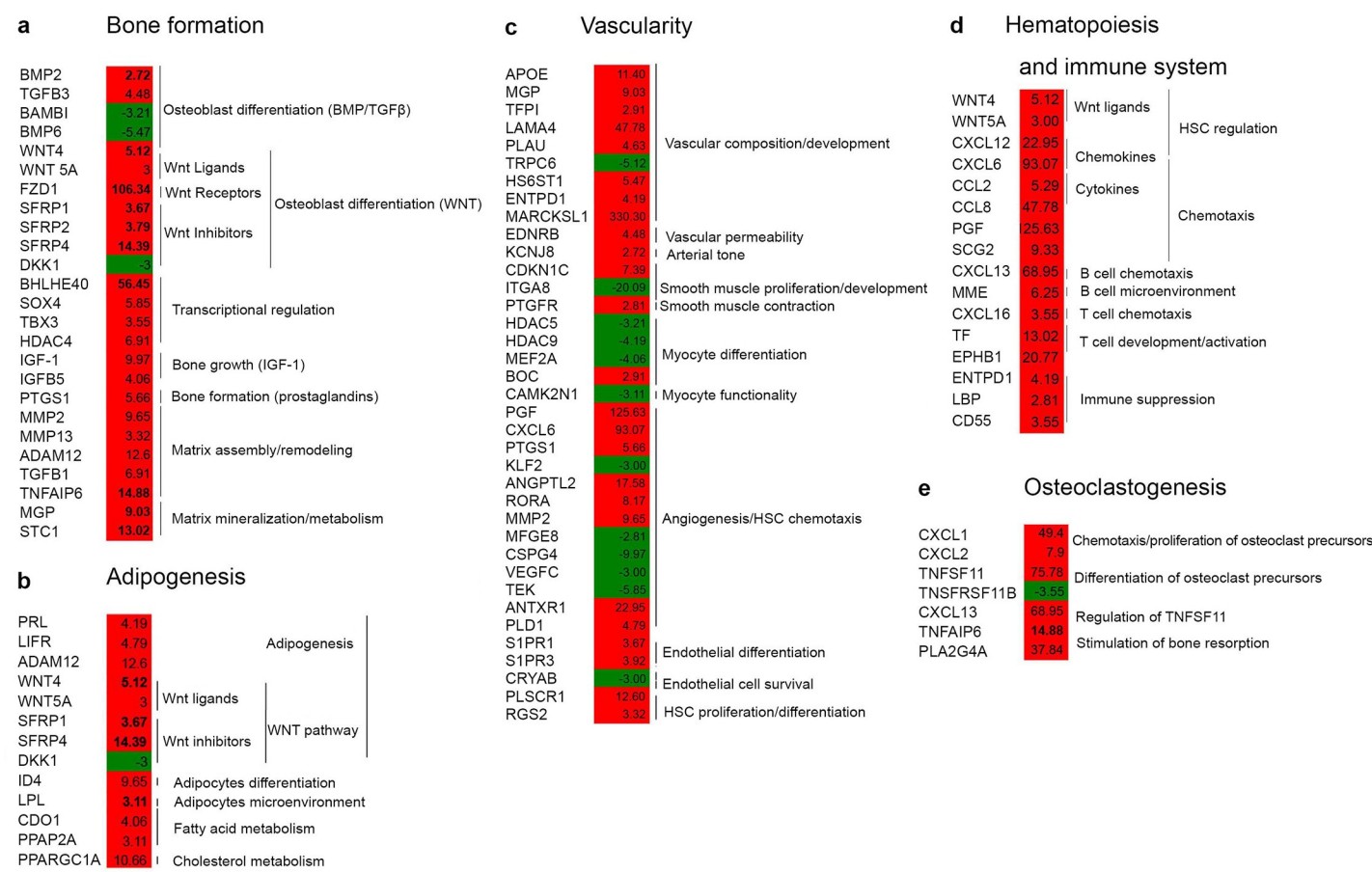

**Fig 5. Modulated genes organized according to hBMSC *cis* and *trans* properties.** Modulated genes were analyzed by IPA, and the top ten physiological functions and the relative genes given by IPA core analysis (S1 Fig) were clustered into groups related to fundamental tissue changes in FD, taking into account hBMSC autocrine and paracrine properties (see also Fig 7). (A) Osteogenesis; (B) Adipogenesis; (C) Vascularity; (D) Hematopoiesis and immune regulation; (E) Osteoclastogenesis. Over-expressed genes are indicated in red, under-expressed genes in green.

metabolic factors, *Phosphatidic Acid Phosphatase 2a* (*PPAP2A*), a negative regulator of insulin signaling [38] and of lipolysis [39] were also upregulated in Gsα$^{R201C}$ hBMSCs. These modulations are typical of mature adipocytes suggesting an activation of adipocytic differentiation of Gsα$^{R201C}$. However both in FD lesions and in Gsα$^{R201C}$-expressing mice, there is a lack of adipocytes [11, 16]. These data taken together suggest a complex, non-linear, interplay of factors involved in adipogenesis in the FD condition. *Prolactin* (*PRL*), a factor stimulated by cAMP activators [40] was also induced. PRL promotes early pre-adipocyte differentiation [41] via the activation of the adipogenic transcription factors such as CCAAT/Enhancer Binding Protein Beta (C/EBPB) and Peroxisome Proliferator Activated Receptor Gamm (PPARG) [42, 43]. The C/EBPB/PPARG route of adipogenic modulation is also stimulated by LIF Receptor Alpha (LIFR), which was upregulated in Gsα$^{R201C}$ hBMSCs [44]. Although *C/EBPB* and *PPARG* were not directly found to be significantly upregulated, *PPARG Coactivator 1 Alpha* (*PPARGC1A*), a transcriptional co-activator of PPARG [45], was overexpressed by Gsα$^{R201C}$. As indicated previously, Gsα$^{R201C}$ induced the WNT non-canonical pathways. This leads to enhanced osteoblastogenesis, but also to inhibition of adipogenesis [46]. Taken together, these results suggest a complex alteration of the adipogenic program in Gsα$^{R201C}$ transduced hBMSCs.

## Gsα$^{R201C}$ hBMSCs and browning of fat

Adipocytes are subdivided into different classes including white adipocytes, brown adipocytes, inducible brown adipocytes referred to as "beige" or "brite" adipocytes [47, 48], and marrow adipoctyes appear to represent a fourth class. Excess cAMP, the fundamental downstream effector of Gsα activating mutations observed in Gsα$^{R201C}$ hBMSCs mutant cells [16], is the prime stimulator of beige fat induction. Beige fat develops from specific precursor cells that may include pericyte-like cells [49]. Although limited information is available, bone marrow fat has been noted to exhibit brown fat-like features [50]. Given this notion, we explored whether the abnormal adipocytic differentiation program induced by the FD mutation in hBMSC precursors would push the cells towards a specific fat subtype. We took as a reference the transcriptional markers described for beige fat by Wu and coworkers [51]. We then analyzed the modulation of expression of these genes in Gsα$^{R201C}$ hBMSCs array raw data. This analysis indicated that Gsα$^{R201C}$ hBMSCs displayed an expression profile with numerous elements that overlap with the beige fat profile, including the overexpression of *CEBPA*, *Insulin Like Growth Factor 1 Receptor* (*IGF1R*) and *PPARGC1A* (Fig 6). We believe that this preliminary information suggests that the FD mutation could be involved in browning of white fat and that this process deserves further investigation in the FD phenotype.

## Vascularity in Gsα$^{R201C}$ hBMSCs

A high number of genes correlated to vascularization in Gsα$^{R201C}$ hBMSCs (Fig 5C), suggesting that the mutation drives alteration of angiogenic processes through the activity of BM skeletal progenitors. Among the modulated genes, we observed high expression of *Angiopoietin-Like 2* (*ANGPTL2*). This gene encodes for a factor belonging to the angiopoietin-like family of proteins, which are structurally related to angiopoietins but do not bind to Angiopoietin's receptor, TIE-2. This family of proteins has been linked to neoplastic transformation and metabolic disease [52]. In addition, it has been shown that ANGPTL2 may be involved in chronic inflammatory conditions [53]. A second pro-angiogenic factor modulated in Gsα$^{R201C}$ hBMSCs was *PGF* (*Placental Growth Factor*) (Fig 7C), a member of the family of factors controlling vascular endothelium formation normally observed at an embryonic stage. It is also associated with pathological angiogenesis, through a phospholipase C-dependent pathway, and protein kinase C via the activity of MAP kinases [54]. We also observed that Gsα$^{R201C}$ induced *Sphingosine-1 Phosphate Receptor 1* (*S1PR1*) and *3* (*S1PR3*), two G protein coupled receptors important for downstream stabilization of new vessels [55, 56]. Along with these factors, Gsα$^{R201C}$ induced the chemokine gene, *C-X-C Motif Chemokine Ligand 6* (*CXCL6*), which acts as a chemoattractant for endothelial cells during angiogenesis [57]. On the other hand, the under-expression of *Krüppel Like Factor 2* (*KLF2*) by Gsα$^{R201C}$ mutant cells further underlines the proangiogenic effect of the mutation, since this is a factor identified as a potent inhibitor of angiogenesis through the negative regulation of Hypoxia inducible Factor 1 Alpha Subunit (HIF-1α), downstream of the proangiogenic factors, such as Interleukin 8 (IL-8), Angiopoietin 2 (ANG2), and Vascular Endothelial Growth Factor (VEGF) [58].

## Hematological and immune system in Gsα$^{R201C}$ hBMSCs

The physical and molecular support of hBMSCs is essential to promote the differentiation of cells in the hematopoietic lineage [14]. Genes belonging to the hematological and immune systems were modulated by Gsα$^{R201C}$ mutation (Fig 5D), including factors involved in cell proliferation (e.g., *C-C Motif Chemokine Ligand 2* (*CCL2*) [59]), and in the balance between cell survival and apoptotic processes (e.g., *Transferrin* (*TF*) [60], *TNF Superfamily Member 11* (*TNFSF11*) [61]). The upregulation of genes with proliferation and anti-apoptotic activities

| Beige fat marker | GsαR201C hBMSCs |
|---|---|
| ACTRIIB | 0.57 |
| ATG7 | 0.71 |
| C/EBPA | 2.36 |
| CIDEA | 0.58 |
| ELOVl3 | 2.67 |
| IGF1R | 32.96 |
| IR | 1.40 |
| NG2 | 1.44 |
| PPRGC1A | 10.10 |
| PLAC8 | 2.04 |
| PRLR | 5.99 |
| PTEN | 1.52 |
| SRC | 3.82 |
| TRPM8 | 2.70 |

**Fig 6. In silico comparative analysis of beige fat properties and Gsα[R201C] induced modulation.** Taking as a reference the transcription status of beige fat markers ([51], Beige fat, left column), we observed that Gsα[R201C] hBMSCs displayed an overlapping profile (Gsα[R201C] hBMSCs, right column). Over-expressed genes are indicated in red, under-expressed genes in green. Fold change observed in Gsα[R201C] hBMSCs is indicated.

suggest survival of transduced cells, but is presumably an acute early reaction to the presence of the mutation, since an increase in apoptosis is observed in FD lesions [12]. Cytokines and chemokines were also upregulated by Gsα[R201C] (Fig 5D). This suggests a stimulating/signaling activity involved in the chemotaxis of cells of the myeloid lineage (erythrocytes, monocytes, basophils, eosinophils, neutrophils) and lymphocyte lineage (B and T cells). The activation of chemotaxis proposes that Gsα[R201C] hBMSCs could be involved in the activation of immune

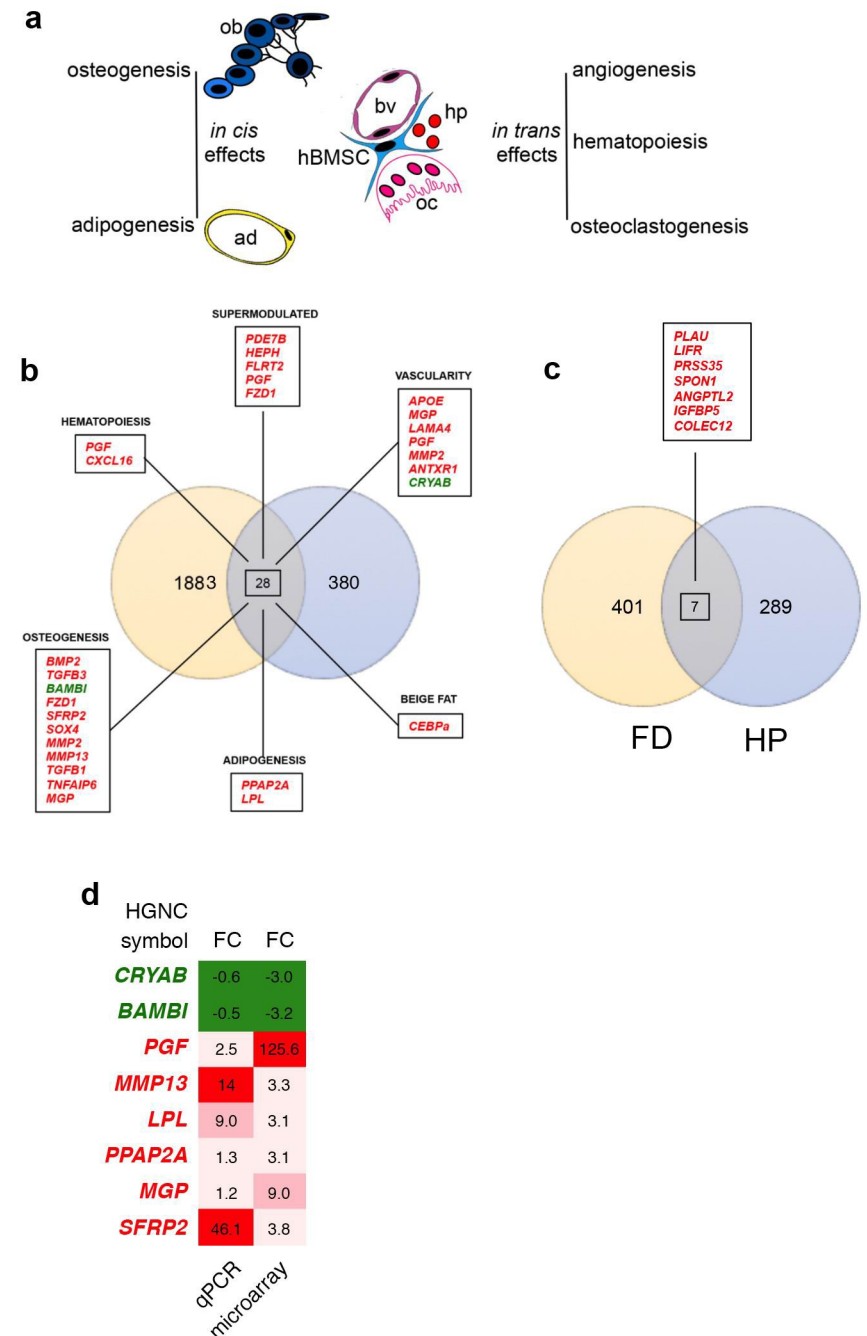

**Fig 7. hBMSCs and FD-related processes.** (A) Cartoon depicting the *in cis* differentiation properties of BMSCs into osteoblasts and adipocytes, along with the *in trans* effects of BMSCs on hematopoiesis, osteoclastogenesis and vascularity (ad, adipocyte; ob osteoblast, oc osteoclast. Hp, hematopoietic cells, bv blood vessel). (B) Venn diagram of differentially expressed genes (under-expressed, in green; over-expressed, in red) identified by Zhou et al. [10] on human FD craniofacial samples and those from Gsα[R201C] hBMSCs microarray analysis described in this work. 28 genes were commonly regulated and those related to specific *trans* and *cis* FD related alterations are indicated. Red over-expressed genes, green under-expressed genes. (C) Venn diagram of commonly expressed genes (over-expressed, in red) by comparing data from this work to PTH treated samples described by [68, 69]. (D) Fold change comparison of differentially expressed genes, analyzed by microarray and qPCR validated. Genes were chosen among the commonly regulated and those related to specific *trans* and *cis* FD related alterations.

cell trafficking, humoral and cell-mediated immune responses and inflammatory phenomena in FD. Also in this case, presumably this is an acute early reaction to the presence of the mutation, since in FD lesions, which are the results of chronic expression of Gsα$^{R201C}$ mutation, lack of hematopoiesis and absence of inflammation was observed [1, 3].

## Stimulated osteoclastogenesis in Gsα$^{R201C}$ mutant hBMSCs

There is a tight connection between cells in the hematopoietic lineage and bone remodeling, and many chemokines involved in inflammatory responses are also responsible for the regulation of osteoclastogenesis. *C-X-C Ligands* (*CXCLs*) such as *CXCL1*, *CXCL13* and *CXCL2* are highly over expressed in Gsα$^{R201C}$ mutant hBMSCs (Fig 5B), are able to attract osteoclast precursors to the bone environment [62], to enhance the proliferation of osteoclast precursor cells of bone marrow-derived macrophages [63] and to stimulate *Tumor Necrosis Factor Ligand Superfamily member 11* (*TNFSF11*, also known as *RANKL*) expression [64], which has recently been shown to be over-expressed in human FD and in transgenic models of the disease [20, 65]. *Prostaglandin E2* (*PGE2*) produced by osteoblasts is also a potent stimulator of bone resorption, and *Phospholipase A2 Group IVA* (*PLA2G4A*), highly expressed in our system, plays a key role in *PGE2* production [66]. Altogether, these data suggest strong osteoclast activation, consistent with the pronounced osteoclastogenesis observed in FD lesions [67] and in the FD mouse model [11].

## Conclusions and comparison to FD samples

Multiple lines of evidence point to the complexity of the pathophysiology of FD, in which multiple factors have to be taken into account for understanding the disease and for designing successful therapeutic strategies. The overall FD phenotype can be attributed to alterations of skeletal stem/progenitors that develop into osteogenic or adipogenic cells (*in cis*), which are also known to provide support to angiogenesis, hematopoiesis, and osteoclastogenesis (*in trans*) (Fig 7A). We thus reasoned that the analyses of these processes upon introduction of the FD mutation would offer a new perspective view of the development of the disease. Our data suggest that the FD mutation profoundly alters the properties of skeletal stem/progenitor cells by directing hBMSCs towards formation of disorganized bone with a concomitant altered fat development. In addition, the mutation created an altered *in trans* environment that diverted the overall system towards neovascularization and osteoclastogenesis.

It will be interesting to analyze in vivo how these observations correspond to the evolution of the disease. In fact, we have already observed that specific aspects that were prospectively defined through the data presented here were also observed in vivo. For example, increased osteoclastogenesis is known to occur in human FD tissue and we observed that the same effect was elicited by the mutation in the animal model of FD, and this could represent a targetable pathway of treatment for FD patients [11]. The comparison with human samples will also be of interest, however, keeping in mind a caveat based on the fact that studies on human FD have been performed on samples obtained when the disease stage is overt. One such example is the array analysis performed by Zhou and co-workers on craniofacial samples obtained from FD patients. In particular, it was interesting to observe that some aspects that we observed on skeletal stem/progenitors were also present in FD samples (Fig 7B). ADAM proteins and other matrix-related factors were indeed upregulated in both experimental systems. Along with this, in both systems *Phosphodiesterase 7B* (*PDE7B*) was over-expressed, which can be considered as a buffering process, activated to compensate for excess cAMP as we already observed in a previous study in human skeletal progenitors [16]. Moreover, we want to highlight the over-expression of *CEBPs* in both systems, factors related to browning of white fat.

Given the link between parathyroid hormone and the skeletal phenotype, we made a comparative analysis between our microarray data and those described by Li and co-workers on rat samples treated with parathyroid hormone (PTH) [68, 69]. To compare the expression profiles we considered our microarray data for FD and the list of modulated genes corresponding to human orthologues of rat regulated genes identified by Li and co-workers. This analysis highlighted 7 genes which were over-expressed in both FD and PTH samples, including *PLAU* and *ANGPTL2*, which can be associated with vascularity alterations, *IGFBP5* with bone formation, and *LIFR* correlated with adipogenesis (Figs 7C and 5).

Finally, we tested differentially expressed genes by qPCR for validation (Fig 7D and S2 Fig). The genes were chosen among the commonly regulated and those related to specific *trans* and *cis* FD related alterations. Eight transcripts, corresponding to 6 upregulated and 2 downregulated genes in the microarray analysis, were tested by qPCR. Tested genes were qualitatively modulated as on microarray.

This is the first analysis of skeletal stem/progenitor cell responses to introduction of the FD mutation and we believe it provides a useful background for further studies on the molecular basis of the disease and for the identification of novel potential therapeutic targets.

## Materials and methods

### hBMSCs

hBMSCs were cultured in α-MEM, 2mM L-glutamine, 20% FBS, 100U/ml penicillin, 100μg/ml streptomycin (Invitrogen, Carlsbad, CA) [13, 21–25]. 293T cells (ATCC CRL-11268) were used for viral production and maintained in DMEM (Invitrogen), 10% FBS, 100U/ml penicillin and 100μg/ml streptomycin.

### Vectors

Lentiviral vectors (LV-GSα$^{R201C}$ and LV-ctr) were constructed, produced and titrated as previously described [16]. The LV-vector integrated copy number was calculated by qPCR as described in [16] and was established as ~1 copy of integrated lentiviral sequence per transduced cell. hBMSCs were transduced with LV-GSα$^{R201C}$ and LV-ctr or mock treated as previously described [16].

### Western blotting

For Western blotting, ten days following infection protein extracts were obtained as previously described [70]. Filters were then incubated with anti-Gsα (sc-823 Santa Cruz Biotechnology), anti-actin-HRP conjugated (sc-1615, Santa Cruz Biotechnology) and anti-HA (sc-805 Santa Cruz Biotechnology) antibodies. Anti-rabbit HRP-conjugated (sc-2357, Santa Cruz Biotechnology) has been used as secondary antibody.

### qPCR, gene expression profiling and data analysis

Total cellular RNA was isolated from the cell populations using an RNeasy RNA isolation kit (Qiagen) as described in [71]. Disposable RNA chips (Agilent RNA 6000 Nano LabChip kit) were used to determine the concentration and purity/integrity of RNA samples using an Agilent 2100 bioanalyzer. cDNA synthesis, biotin-labeled target synthesis, HG-U133 plus 2.0 GeneChip (Affymetrix, Santa Clara, CA) array hybridization, staining and scanning were performed as described in [71, 72]. The amount of a transcript mRNA (signal) was determined by the Affymetrix GeneChip Operative Software (GCOS) 1.2 absolute analysis algorithm [73]. All expression values for the genes in the GCOS absolute analyses were determined using the

global scaling option. Alternatively, probe level data were converted to expression values using the Robust Multiarray Average (RMA) procedure [72, 74] or dChip procedure (invariant set) [72, 75]. Data were then filtered and analyzed using dChip®, Partek GS® and R (Bioconductor). In particular, the R-AffyQC Report, R-Affy-PLM, R-RNA Degradation Plot and dChip QC were used to perform all quality controls. Under- and over-expressed genes were obtained merging the overlapping genes coming from three approaches: 1) Paired pair-wise comparisons, between the LV-Gsα^R201C transduced hBMSCs and the two controls, mock and LV-ctr-transduced hBMSCs, were performed using the Affymetrix GCOS comparison algorithm and identifying probe sets showing an "increased" or "decreased" call and a signal log ratio greater than 1 or less than -1 in comparison of all replicates of only LV-Gsα^R201C transduced hBMSCs vs mock. 2) Alternatively, differentially expressed genes were obtained using the dChip Compare Sample procedure. Briefly, the comparison criteria utilized in dChip requires the fold change and the absolute difference between the paired sample means to exceed user defined thresholds (in the present study, 2 and 200 respectively). The "Use lower 90% confidence bound" was selected to use the lower confidence bound of fold changes for filtering. The lower confidence limit is intended as a conservative estimate of the real underlying fold change. False Discovery Rate (FDR) was used to adjust p-values for multiple comparisons by 100 random permutations of the group labels. 3) Finally, a paired t-test was performed between transduced and untreated groups using Partek GS® selecting genes with a p-value less than 0.01 and a contrast greater than 2-fold change. Partek GS® was also used to manage the gene lists coming from different analysis. The data set containing the Affymetrix probe identifiers selected as under- and over-expressed in LV-Gsα^R201C transduced hBMSCs was uploaded into EnrichR (http://amp.pharm.mssm.edu/Enrichr/) and Ingenuity Pathway Analysis (www.ingenuity.com). Enrichment analysis of under- and over-expressed genes was performed with the publicly available tool, EnrichR (http://amp.pharm.mssm.edu/Enrichr), that provides access to various gene-set libraries [76, 77]. EnrichR currently contains annotated gene sets from 102 gene set libraries organized in 8 categories. We considered pathways and Gene Ontology terms as enriched if their p-value was lower than 0.05 and ranked them Combined Score (CS). The CS is a combination of the p-value and z-score calculated by multiplying the two scores as follows: CS = log(p)*z where p is the p-value computed using Fisher's exact test, and z is the z-score computed to assess the deviation from the expected rank.

qPCR were performed as described in [70] using the following primers:
```
CRYAB F: 5'-CAGAGGAACTCAAAGTTAAGG
CRYAB R: 5'-ATGAAACCATGTTCATCCTG
LPL F: 5'-ACACAGAGGTAGATATTGGAG
LPL R: 5'-CTTTTTCTGAGTCTCTCCTG
MGP F: 5'-ATAAAAACCTCACAGCCTTC
MGP R: 5'-CCATAACACAAAGTTACTACCG
PGF F: 5'-AGCTCCTAAAGATCCGTTC
PGF R: 5'-GACGGTAATAAATACACGAGC
SFRP2 F: 5'-GACCTAGACGAGACCATC
SFRP2 R: 5'-ATACCTTTGGAGCTTCCTC
PPAP2A F: 5'-CACTTTATCTTCAAGCCAGG
PPAP2A R: 5'-ACTAATATTGCAACCAGAGC
BAMBI F: 5'-AAGGTGAAATTCGATGCTAC
BAMBI R: 5'-TCAAGAAGTCTAGAGAAGCAG
MMP13: 5'-AGGCTACAACTTGTTTCTTG
MMP13: 5'-AGGTGTAGATAGGAAACATGAG
```

qPCR reactions were performed with the Applied Biosystems PRISM 7300 Real Time PCR System with QuantiTect SYBR Green PCR Kit. To obtain quantification with respect to mock cells, quantification cycle values (Cq) were exported and ansalysed with Excel, the data were calculated with the $2^{-\Delta\Delta Cq}$ method as described in [70]. LV-Gsα$^{R201C}$ treated hBMSCs relative fold change were then subtracted to mock relative fold change. Data are reported as means of duplicates or more data obtained from two independent biological samples. For qPCR analysis p is the p-value computed using Student's $t$ test ($^*p < .05$, $^{**}p < .01$ $^{***}p < .001$).

## Ethics approval

Established human bone marrow stromal cell line (hBMSCs) were obtained from healthy donors with informed consent per institutionally approved protocols by the Ethical Committee of Sapienza University of Rome, Policlinico Umberto I venue (Ref. 5313, version n 2.0–26.04.19) [13, 21–25].

## List of abbreviations

Fibrous dysplasia (**FD**); G protein-coupled receptor complex (**Gs**α); Gsα mutation involving a substitution of arginine at position 201 by histidine or cysteine (**Gsα$^{R201H\ or\ R201C}$**); McCune-Albright syndrome (**MAS**); bone marrow stromal cells (**BMSCs**); skeletal stem cells (**SSCs**); lentiviral vector expressing the constitutively active form of the Gsα protein carrying the R201C mutation (**LV-Gsα$^{R201C}$**); lentiviral vector with control vector (**LV-ctr**); untransduced (**mock-treated**) controls; R201C mutation transduced cells (**R**), mock-treated cells (**M**), cells transduced with LV-ctr (**C**); Gene Ontology (**GO**); Enrichr's Combined Score (**ECS**); Kyoto Encyclopedia of Genes and Genomes (**KEGG**); Affymetrix GeneChip Operative Software (**GCOS**); Robust Multiarray Average (**RMA**); False Discovery Rate (**FDR**).

A Disintegrin and Metalloprotease (**ADAM**); Phosphodiesterase 7B (**PDE7B**); CCAAT-Enhancer Binding Proteins (**CEBPs**); adenylyl cyclase (**AC**); Protein Kinase A (**PKA**); activator protein 1 (**AP-1**); Interleukin 6 (**IL-6**); Platelet-derived growth factor β (**PDGFβ**); Bone morphogenetic protein (**BMP**); ATPase Sarcoplasmic/Endoplasmic Reticulum Ca2+ Transporting 2 (**ATP2A2**); Transforming Growth Factor-ß (**TGFß**); Wingless-Type MMTV Integration Site (**WNT**); Ingenuity Pathway Analysis (**IPA**);Frizzled Class Receptor 1 (**FZD1**); Secreted Frizzled Related Proteins (**SFRPs**); Matrix Gla Protein (**MGP**); Stanniocalcin-1 (**STC1**); Matrix Metalloproteinases (**MMPs**); A Disintegrin and Metalloproteinase Domain 12 (**ADAM12**); A Disintegrin and Metalloproteinase Domain with Thrombospondin Type 1 Motif 2 (**ADAMTS2**); T-Box Protein 3 (**TBX3**); Lipoprotein Lipase (**LPL**); Phosphatidic Acid Phosphatase 2a (**PPAP2A**); Prolactin (**PRL**); CCAAT/Enhancer Binding Protein Beta (**C/EBPB**); Peroxisome Proliferator Activated Receptor Gamm (**PPARG**); LIF Receptor Alpha (**LIFR**); PPARG Coactivator 1 Alpha (**PPARGC1A**); Insulin Like Growth Factor 1 Receptor (**IGF1R**); Angiopoietin-Like 2 (**ANGPTL2**); MAP kinases (**MAPK**); Sphingosine-1 Phosphate Receptor 1 (**S1PRs**); C-X-C Motif Chemokine Ligand 6 (**CXCL6**); Krüppel Like Factor 2 (**KLF2**); Hypoxia inducible Factor 1 Alpha Subunit (**HIF-1**α); Interleukin 8 (**IL-8**); Angiopoietin 2 (**ANG2**); Vascular Endothelial Growth Factor (**VEGF**); C-C Motif Chemokine Ligand 2 (**CCL2**); Transferrin (**TF**); C-X-C Ligands (**CXCLs**); Tumor Necrosis Factor Ligand Superfamily member 11 (**TNFSF11**, also known as **RANKL**); Prostaglandin E2 (**PGE2**); Phospholipase A2 Group IVA (**PLA2G4A**).

## Availability of data and material

The data discussed in this manuscript have been deposited in NCBI's Gene Expression Omnibus [78] and are accessible through GEO Series accession number **GSE109818** (https://www.ncbi.nlm.nih.gov/geo/query/acc.cgi?acc=GSE109818).

## Supporting information

**S1 Table. Under- and over-expressed genes were obtained merging the overlapping genes coming from three approaches: 1) paired pair-wise comparisons using the Affymetrix GCOS comparison algorithm, 2) using dChip Compare Sample procedure, 3) paired t-test using Partek GS®.** FC, fold change. T test, paired t-test applied comparing LV-Gsα$^{R201C}$ to mock treated hBMSCs.
(DOCX)

**S1 Fig. In silico analysis.** (A) Flow chart of the different steps applied for the functional and statistical analysis of array data. (B) Functional pathways were evaluated in the Gsα$^{R201C}$ data set by Ingeuity Pathwaay Analysis (IPA). Significant pathways were defined by assessing the number of molecules mapping to the pathway and by Fisher's exact test calculated p-value. Top ten scoring pathways are showed in the histogram.
(TIF)

**S2 Fig. qPCR analysis.** (A) qPCR analysis on cDNA extracted from LV-Gsα$^{R201C}$ and mock treated hBMSCs, for the different indicated genes. Data are obtained from tripilicate measurements of independent biological duplicates shown as the difference of LV-Gsα$^{R201C}$ fold change respect to mock treated samples, each dot represent an individual sample. *P* values were calculated with Student's *t* test ($^{*}p < .05$, $^{**}p < .01$ $^{***}p < .001$).
(TIFF)

## Acknowledgments

We thank S. Piersanti for her technical contribution. This work is in the memory of P. Bianco.

## Author Contributions

**Conceptualization:** Mara Riminucci, Isabella Saggio.

**Data curation:** Domenico Raimondo, Cristina Remoli, Letizia Astrologo, Romina Burla, Mattia La Torre, Fiammetta Vernì, Enrico Tagliafico, Alessandro Corsi, Simona Del Giudice, Agnese Persichetti, Giuseppe Giannicola, Isabella Saggio.

**Formal analysis:** Domenico Raimondo, Cristina Remoli.

**Investigation:** Pamela G. Robey.

**Writing – original draft:** Pamela G. Robey, Mara Riminucci, Isabella Saggio.

**Writing – review & editing:** Pamela G. Robey, Mara Riminucci, Isabella Saggio.

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
