## [Decision Letter · Decision Letter 0]

17 Oct 2019

PONE-D-19-26235

Changes in gene expression in human skeletal stem cells transduced with constitutively active Gsα correlates with hallmark histopathological changes seen in fibrous dysplastic bone

PLOS ONE

Dear Dr. Saggio,

Thank you for submitting your manuscript to PLOS ONE. After careful consideration, we feel that it has merit but does not fully meet PLOS ONE’s publication criteria as it currently stands. Therefore, we invite you to submit a revised version of the manuscript that addresses the points raised during the review process.

This study has potentially valuable information to the field of fibrous dysplasia.  However, it is critical to address the concerns raised by the reviewers with respect to the rigor of the study, i.e. R201C mutation in all 3 donors and the protein expression levels.  In addition, please provide the rationale of the cell cultures and the comparison/contrast with hyperparathyroidism.   

We would appreciate receiving your revised manuscript by Dec 01 2019 11:59PM. To enhance the reproducibility of your results, we recommend that if applicable you deposit your laboratory protocols in protocols.io, where a protocol can be assigned its own identifier (DOI) such that it can be cited independently in the future. For instructions see: http://journals.plos.org/plosone/s/submission-guidelines#loc-laboratory-protocols

We look forward to receiving your revised manuscript.

Kind regards,

Yin Tintut, Ph.D.

Academic Editor

PLOS ONE

Journal Requirements:

2. Our internal editors have evaluated your manuscript and determined that it is within the scope of our Stem Cell Plasticity in Tissue Repair and Regeneration Call for Papers. This collection of papers is headed by a team of Guest Editors for PLOS ONE and will encompass a diverse range of research articles. Additional information can be found on our announcement page: (https://collections.plos.org/s/stem-cell). If you would like your manuscript to be considered for this collection, please let us know in your cover letter and we will ensure that your paper is treated as if you were responding to this call. If you would prefer to remove your manuscript from collection consideration, please specify this in the cover letter.

Reviewers' comments:

Reviewer's Responses to Questions

**Comments to the Author**

1. Is the manuscript technically sound, and do the data support the conclusions?

Reviewer #1: Yes

Reviewer #2: Yes

2. Has the statistical analysis been performed appropriately and rigorously? 

Reviewer #1: Yes

Reviewer #2: Yes

3. Have the authors made all data underlying the findings in their manuscript fully available?

Reviewer #1: Yes

Reviewer #2: Yes

4. Is the manuscript presented in an intelligible fashion and written in standard English?

Reviewer #1: Yes

Reviewer #2: Yes

5. Review Comments to the Author

Reviewer #1: This manuscript describes gene expression changes (assessed by microarrays) in human bone marrow stromal cells in response to introduction of the FD-associated Gs,alpha R201C mutation. This paper is well-written and contains a systematic and careful discussion about the complex nature of bone disease in FD. THis new dataset will be of some value to researchers studying FD and other cAMP-regulated bone conditions. Overall, these microarray data are clearly described and robust.

Minor comments:

1. In Figure 1A, the three samples transduced with the R201C mutant are all listed as D02R. Is this correct? Or should this instead state D01R, D02R, and D03R? Ideally the authors would perform this analysis using three separate BMSC donors.

2. It would be useful to show an immunoblot demonstrating expression levels of the R201C mutant compared to endogenous Gs,alpha protein levels.

3. The authors could compare/contrast the genes regulated by Gs,alpha R201C expression and genes whose expression is regulated by continuous hyperparathyroidism (PMID 17690103).

Reviewer #2: The manuscript by Raimondo et al., reports transcriptome analysis of human bone marrow stromal cells constitutively expressing a mutated dominant form of the GNAS gene coding for a G-protein coupled receptor. The mutation is a substitution of Arginine to Cysteine at position 201 of the alpha subunit and causes fibrous dysplasia of bone. The cells are transduced with a lentiviral vector carrying the GsαR201C mutation. Previous work by the group has documented that GsαR201C transduced cells manifest the diseased phenotype in vitro and after ectopic in vivo transplantation into mice. In this report the authors concentrate on the expression profile of transduced cells with the aim to find early perturbations in gene expression that may provide potential targets for development of therapeutic strategies. For this, 3 different human samples isolated from healthy individuals were transduced with the constitutive active GsαR201C or with control vectors and expression profile analysis was performed after 15 days. The bioinformatics analysis of the data was directed towards the identification of pathways that may relate to disease phenotype. Interestingly genes belonging to the Wnt pathway as well as genes related to osteoblastogenesis were dysregulated in GsαR201C transduced cells and provide a link to the bone malformation situation observed in fibrous dysplasia (FD) patients. Similarly, an altered adipogenic program is related the block of adipogenesis in FD patients. Interestingly, comparison of the expression profile in this manuscript to published data from FD patients revealed common sets of dysregulated genes. Despite that the expression of GsαR201C is regulated from an exogenous promoter and not from the endogenous locus, the transcriptomic data are informative and can contribute to a better understanding of the molecular pathways involved in FD disease.

Minor comments

1. Please describe better the culture requirements for hBMSCs in the “Material and Methods” part. A far as I know, dexamethasone and ascorbic acid are used for osteogenic differentiation and not for maintaining hBMSCs. Further, you should mention why you culture 293T cells (virus production?).

2. KLF2 is Krüppel-like Factor 2 and NOT Krüpple

6. PLOS authors have the option to publish the peer review history of their article (what does this mean?). If published, this will include your full peer review and any attached files.

Reviewer #1: No

Reviewer #2: Yes: Konstantinos Anastassiadis

---

## [Author Response · Author response to Decision Letter 0]

3 Dec 2019

November 26, 2019

PONE-D-19-26235 

Dear Editor,

We enclose the revised version of the manuscript “Changes in gene expression in human skeletal stem cells transduced with constitutively active Gsα correlates with hallmark histopathological changes seen in fibrous dysplastic bone” by Domenico Raimondo and coworkers.

We hope that you will find this revised version suited for publication in PLOS One.

Thank you for your consideration,

Isabella Saggio

isabella.saggio@uniroma1.it, 

 

\f

PONE-D-19-26235 - REBUTTAL LETTER

Reviewer #1:

This manuscript describes gene expression changes (assessed by microarrays) in human bone marrow stromal cells in response to introduction of the FD-associated Gs,alpha R201C mutation. This paper is well-written and contains a systematic and careful discussion about the complex nature of bone disease in FD. THis new dataset will be of some value to researchers studying FD and other cAMP-regulated bone conditions. Overall, these microarray data are clearly described and robust. 

We thank for the favourable comments.

Minor comments: 1. In Figure 1A, the three samples transduced with the R201C mutant are all listed as D02R. Is this correct? Or should this instead state D01R, D02R, and D03R? Ideally the authors would perform this analysis using three separate BMSC donors.

The analysis was indeed performed on three donors and the Figure was corrected.

 2. It would be useful to show an immunoblot demonstrating expression levels of the R201C mutant compared to endogenous Gs,alpha protein levels.

This experiment was performed and added as Figure 1A.

 3. The authors could compare/contrast the genes regulated by Gs,alpha R201C expression and genes whose expression is regulated by continuous hyperparathyroidism (PMID 17690103).

This comparison was performed and added as Figure 7C.

 

Reviewer #2: 

The manuscript by Raimondo et al., reports transcriptome analysis of human bone marrow stromal cells constitutively expressing a mutated dominant form of the GNAS gene coding for a G-protein coupled receptor. The mutation is a substitution of Arginine to Cysteine at position 201 of the alpha subunit and causes fibrous dysplasia of bone. The cells are transduced with a lentiviral vector carrying the GsαR201C mutation. Previous work by the group has documented that GsαR201C transduced cells manifest the diseased phenotype in vitro and after ectopic in vivo transplantation into mice. In this report the authors concentrate on the expression profile of transduced cells with the aim to find early perturbations in gene expression that may provide potential targets for development of therapeutic strategies. For this, 3 different human samples isolated from healthy individuals were transduced with the constitutive active GsαR201C or with control vectors and expression profile analysis was performed after 15 days. The bioinformatics analysis of the data was directed towards the identification of pathways that may relate to disease phenotype. Interestingly genes belonging to the Wnt pathway as well as genes related to osteoblastogenesis were dysregulated in GsαR201C transduced cells and provide a link to the bone malformation situation observed in fibrous dysplasia (FD) patients. Similarly, an altered adipogenic program is related the block of adipogenesis in FD patients. Interestingly, comparison of the expression profile in this manuscript to published data from FD patients revealed common sets of dysregulated genes. Despite that the expression of GsαR201C is regulated from an exogenous promoter and not from the endogenous locus, the transcriptomic data are informative and can contribute to a better understanding of the molecular pathways involved in FD disease.

  Minor comments  

1. Please describe better the culture requirements for hBMSCs in the “Material and Methods” part. A far as I know, dexamethasone and ascorbic acid are used for osteogenic differentiation and not for maintaining hBMSCs. Further, you should mention why you culture 293T cells (virus production?).

293T cells were used only for virus production and this was indicated in the Materials and Methods section. The protocol for hBMSCs was corrected, there was a mistake in the text and in reference positioning.  

2. KLF2 is Krüppel-like Factor 2 and NOT Krüpple

This was corrected in the text.

---

## [Decision Letter · Decision Letter 1]

17 Dec 2019

Changes in gene expression in human skeletal stem cells transduced with constitutively active Gsα correlates with hallmark histopathological changes seen in fibrous dysplastic bone

PONE-D-19-26235R1

Dear Dr. Saggio,

We are pleased to inform you that your manuscript has been judged scientifically suitable for publication and will be formally accepted for publication once it complies with all outstanding technical requirements.

With kind regards,

Yin Tintut, Ph.D.

Academic Editor

PLOS ONE

Additional Editor Comments (optional):

Reviewers' comments:

Reviewer's Responses to Questions

**Comments to the Author**

1. If the authors have adequately addressed your comments raised in a previous round of review and you feel that this manuscript is now acceptable for publication, you may indicate that here to bypass the “Comments to the Author” section, enter your conflict of interest statement in the “Confidential to Editor” section, and submit your "Accept" recommendation.

Reviewer #1: All comments have been addressed

Reviewer #2: All comments have been addressed

2. Is the manuscript technically sound, and do the data support the conclusions?

Reviewer #1: Yes

Reviewer #2: Yes

3. Has the statistical analysis been performed appropriately and rigorously? 

Reviewer #1: Yes

Reviewer #2: Yes

4. Have the authors made all data underlying the findings in their manuscript fully available?

Reviewer #1: Yes

Reviewer #2: Yes

5. Is the manuscript presented in an intelligible fashion and written in standard English?

Reviewer #1: Yes

Reviewer #2: Yes

6. Review Comments to the Author

Reviewer #1: (No Response)

Reviewer #2: (No Response)

7. PLOS authors have the option to publish the peer review history of their article (what does this mean?). If published, this will include your full peer review and any attached files.

Reviewer #1: Yes: Marc Wein

Reviewer #2: No

---

## [Editor Report · Acceptance letter]

6 Jan 2020

PONE-D-19-26235R1 

Changes in gene expression in human skeletal stem cells transduced with constitutively active Gsα correlates with hallmark histopathological changes seen in fibrous dysplastic bone 

Dear Dr. Saggio:

I am pleased to inform you that your manuscript has been deemed suitable for publication in PLOS ONE. Congratulations! Your manuscript is now with our production department. 

With kind regards,

on behalf of

Dr. Yin Tintut 

Academic Editor

PLOS ONE